# Perceptions of people with Parkinson's and their caregivers of falling and falls-related healthcare services- a qualitative study

**Charlotte L. Owen**[1,2]*, **Christine Gaulton**[3], **Helen C. Roberts**[1,2], **Laura Dennison**[3]

1 Academic Geriatric Medicine, Faculty of Medicine, University of Southampton, Southampton, United Kingdom, 2 National Institute for Health Research Applied Research Collaboration, Wessex, United Kingdom, 3 Centre for Clinical and Community Applications of Health Psychology, School of Psychology, University of Southampton, Southampton, United Kingdom

* c.l.owen@soton.ac.uk

**Data Availability Statement:** Bona fide researchers, subject to registration may request

## Abstract

### Introduction

Falls are common in Parkinson's disease, and a recognised research priority. Falls lead to physical and psychological morbidity in people with Parkinson's disease and their caregivers, however, those with cognitive impairment/ dementia and caregivers have often been excluded from previous studies. This qualitative study explored how people with Parkinson's disease and their family caregivers understood and experienced falling and healthcare services relating to falls prevention and management.

### Methods

A varied and purposive sample of 20 people with Parkinson's disease (40% confirmed or suspected cognitive impairment/ dementia) and 18 caregivers took part in semi-structured interviews. Eight people with Parkinson's disease and their caregivers were interviewed as a dyad, 22 participants were interviewed alone. Interviews were analysed through inductive thematic analysis.

### Results

Four themes were developed: (i) struggling with thoughts and feelings about falling, (ii) recognising and managing risks surrounding falling, (iii) navigating health and care provision for falling, and (iv) changing as a couple due to falling. Different aspects of falls provoked a range of negative emotions and a variety of coping strategies were adopted. Falls and trying to avoid falls burdened a couple in a number of ways; beyond physical health they also affected functioning, physiological wellbeing, and relationships. Dyads analysed falls to understand their aetiology and described working together to manage them. This often happened in the absence of adequate support and advice with little involvement of healthcare professionals. When cognitive impairment/ dementia was present this brought additional challenges to falls management, with caregivers taking on a greater and more frustrating role.

supporting data via University of Southampton repository https://doi.org/10.5258/SOTON/D2329.

**Funding:** The National Institute for Health Research (NIHR) funded this research (https://www.arc-wx.nihr.ac.uk). The views expressed are those of the authors and not necessarily those of the NHS, the NIHR or the Department of Health and Social Care. CO was supported by the University of Southampton NIHR Academic Clinical Fellow (ACF) training programme (https://www.nihr.ac.uk/funding/nihr-academic-clinical-fellowships-in-medicine-2021/25719). CO and HCR were supported by National Institute for Health Research Applied Research Collaboration (NIHR ARC) Wessex (https://www.arc-wx.nihr.ac.uk). HCR was supported by the NIHR Southampton Biomedical Research Centre (https://www.southamptonbrc.nihr.ac.uk). The funders had no role in study design, data collection and analysis, decision to publish, or preparation of the manuscript.

**Competing interests:** The authors have declared that no competing interests exist.

## Conclusion

Dyads required relevant falls-related information and the difficulties associated with cognitive decline should be recognised by researchers and healthcare professionals. Dyads required support in attributing reasons for falls, and increased awareness of healthcare professionals' different roles to improve patient-professional communication and facilitate patient-centred care.

## Introduction

Parkinson's disease (PD) is a progressive neurodegenerative condition, which arises from the selective loss of dopaminergic neurons in the central nervous system and leads to a triad of rigidity, slowness of movement and tremor [1–3]. It is commonly diagnosed when an individual is in their early to mid-60s. Prevalence increases with age; it affects 1% of those over 60 and 3% of those over 80 [1, 4]. The management of PD takes a multidisciplinary approach. Physiotherapists and occupational therapists provide assessment, education, and advice to people with PD (PwPD). Current treatments focus on symptom control, which provide variable benefit [5–8].

Falls are common in PD. Two thirds of PwPD fall each year compared to one third of the general older population [9]. The aetiology of falling in PD is multifactorial; risk factors include freezing of gait and postural instability, with recurrent falls more likely in PwPD with cognitive impairment [10, 11]. Falling often leads to a fear of falling and a decline in physical activity, which is associated with negative outcomes including a decline in both physical function and quality of life [12, 13]. Falls have been identified as a major research priority by PwPD, individuals close to PwPD and healthcare professionals (HCPs) [14].

UK National Institute of Care Excellence (NICE) guidance states that older adults with and without PD who are at risk of falls should undergo an individualised multifactorial risk assessment, be advised of the physical and psychological benefits of reducing falls and offered oral and written information [15]. NICE recommends that PwPD who fall should receive PD specific physiotherapy, although anecdotally PwPD are often referred to generic falls services [16]. Furthermore, systematic reviews of physiotherapy interventions have reported inconclusive results of their ability to reduce fall rate, the proportion of PwPD who fall, fear of falling and quality of life (QOL) [7, 17, 18]. Similarly, studies of occupational therapy interventions for PD, which have not been specific to falls, have produced conflicting results in terms of QOL outcomes [19–21]. Rehabilitative interventions may also be less effective in those with more advanced PD, for whom falling is more common [22].

In older people without PD falls can lead to shock and embarrassment [23, 24]. Individuals may attribute falls due to environmental factors or to personal error, which may help to maintain their personal identity, with falls associated with being old and infirm [23, 25, 26]. Reflecting on a fall and seeking to understand why a fall has occurred is a vital part of falls prevention and management, allowing individuals to instigate strategies to prevent falls and to maintain control [27]. Notably, whilst individuals without a chronic disease may be more likely to describe falls as a normal part of ageing, individuals with a chronic disease may be more likely to attribute falls to their disability [28]. It is important that researchers and clinicians explore how PwPD interpret falls to support the development of interventions to help them.

The vast majority of studies of falls in PwPD have utilised quantitative methodology, with use of standardised quantitative scales to report psychological outcomes such as fear of falling

[13, 29]. Whilst more numerous qualitative studies have been conducted in older people without PD, as falls are more common and complex in PwPD it is important that the experiences of PwPD are explored [9–11]. A recent study reported that PwPD experience fear of falling as a constant disturbance in everyday life, increasing over time as they become more aware of falls and fluctuating in line with the variability of PD symptoms that they experience [30]. In individuals without PD, fear of falling can lead to restriction of activities and reduced social interaction of both the individual who falls and the caregiver [24, 25].

Previous research has highlighted that individuals with and without PD may not discuss falling with HCPs [10, 31]. Falls prevention advice may be disregarded, with individuals perceiving that it is common sense and relates to individuals older and more disabled than themselves [32]. It may also be felt unwarranted amongst individuals who are satisfied with their current situation, or unhelpful in the setting of cognitive impairment [23, 33]. Individuals who fall often receive significant psychosocial and practical support from others [23, 24]. Caregivers of individuals with cognitive impairment, but without PD, have described greater concern about falls than the individual who falls and caregivers of PwPD have reported a need for more information and advice about the management of falls [23, 34].

This study explores how PwPD and their family caregivers understand and experience falling and healthcare services relating to falls prevention and management. It extends on existing literature by exploring the perspectives of people who are particularly affected by falling yet whose experiences and views are currently under-researched [30]. Firstly, we sought to include PwPD with cognitive impairment and dementia. Cognitive impairment is common in PwPD, with mild dysfunction identified in 24% of newly diagnosed PwPD, with prevalence increasing with patient age and disease duration [35, 36]. However, previous falls studies have commonly excluded PwPD with cognitive impairment [37–44]. Secondly, we included the perspectives of caregivers. The majority of PwPD live in their own homes helped by family members, 'informal caregivers', who provide vital physical, emotional and social support [10, 45, 46]. PD leads to transformation of the lives of the PwPD and the caregiver, to include shifts in roles and responsibilities and restrictions in social activities [30, 37, 40, 47, 48]. Caregivers of PwPD often feel unprepared and unsupported in this role, and the onset of falling has been associated with increased caregiver burden [34, 49]. Given caregivers' considerable involvement in the day to day management of PD it is vital that their perspectives are understood and reacted to [34, 50]. Most qualitative studies have explored either PwPD or caregiver perspectives; rarely have both been studied in tandem, however, recent studies have suggested that this is a fruitful approach [37, 47]. A recent systematic review of dyadic challenges and of couples coping with chronically disabling physical and sensory impairments, which included studies of PwPD, people with multiple sclerosis and stroke, highlighted the interpersonal experience of disability [51]. Exploring the experiences of falling in both members of the PwPD/ caregiver dyad is vital given that they are the individuals managing PD day to day. Including caregivers also facilitated the inclusion of PwPD with cognitive impairment/ dementia, who might otherwise have been unable to take part.

We aimed to gain insights into how and when falls occur, the impact of falls, if and how people try to prevent or manage falling and how current healthcare services are experienced.

## Methods

The study adopted a qualitative design using semi-structured interviews as the data collection method. The COREQ checklist (Consolidated criteria for reporting qualitative studies) was followed to ensure comprehensive reporting of the study [52]. A qualitative approach facilitated exploration of the complexities of falling in depth and detail and allowed PwPD and their

caregivers to provide their own perspective, rather than be limited by the priorities, agendas and measures of researchers.

## Participant recruitment

PwPD and caregivers were recruited from Parkinson's UK Support Groups in Hampshire, UK. PwPD were eligible where they had (i) a diagnosis of idiopathic PD, (ii) fallen in the last year and (iii) lived in their own home. Where PwPD met the inclusion criteria for the study, if they had an informal caregiver, both the PwPD and the informal caregiver were invited to take part. When registering interest, eligible participants were given written information about the study, a brief PwPD questionnaire to identify demographic and brief health-related details including a self-report of whether they had diagnosed or suspected cognitive impairment/ dementia. They also completed a measure of fear of falling (short form Falls Self-efficacy Scale-International (FES-I), and caregivers completed a measure of caregiver burden, the Zarit Burden Interview (ZBI) (short version) [53, 54]. Participants were provided with a pre-paid envelope to return the questionnaires to the lead researcher. These measures were used for purposive sampling, and for describing the characteristics of the final sample and understanding and interpreting their data. A purposive sample that varied in terms of demographic characteristics (gender, lived alone/ with caregiver), high and low levels of fear of falling, diagnosed or suspected cognitive impairment/dementia and high and low caregiver burden was identified and potential participants telephoned to arrange an interview. All participants invited to interview agreed to participate.

## Data collection

A semi-structured interview schedule consisting of open questions and probes allowed exploration of participants' experiences and understandings of falling and views about falls management, healthcare provision and resources available to support them (S1 File).

PwPD and caregivers were invited to be interviewed separately. It was acknowledged that participants may not feel comfortable discussing issues, such as caregiver burden, in front of their close friend/ relative. However, participants were able to be interviewed together at their request. Interview questions in single and dyadic interviews were congruous to allow for later combined analysis [55]. Interviews took place between March and July 2017 at the participants' home address by CO (BM, BSc.), a female medical doctor who was employed as a clinical research fellow (n = 16), and CG (BSc.) a female candidate for an MSc. in Health Psychology (n = 13). Interviews were audio-recorded. Field notes were completed by CO and CG and communicated throughout the data collection process.

## Data analysis

Audio recordings were transcribed verbatim and were analysed through thematic analysis supported by QSR NVivo 11 [56, 57]. CO familiarised herself with the data transcripts and assigned descriptive codes to sections of text containing information relating to the research questions. In keeping with reflexive thematic analysis, coding was fluid and interpretative [57]. Coding was undertaken by CO with input from LD to provide additional analytical insight to aid data interpretation. After 18 interviews had been coded, related codes were reviewed and grouped into 'clusters' and evolving subthemes and themes were discussed with LD and HR. Themes were developed inductively and were not considered to be pre-existing entities lying within the data [57].

## Ethics

Ethical approval was granted by the University of Southampton Faculty of Medicine Ethics Committee ERGO reference 29763. Informed written consent was obtained prior to each of the interviews. Where the PwPD was unable to provide informed written consent because of cognitive impairment/ dementia, their caregiver gave consent as a consultee.

## Results

### Participants

38 participants consisting of 20 PwPD and 18 caregivers were interviewed (Tables 1 and 2). Most PwPD were male (70%), median age of PwPD was 72.5 years and median time from diagnosis was 12.8 years. 20% of PwPD (n = 4) had a diagnosis of cognitive impairment/ dementia, a further 20% (n = 4) self-reported cognition/memory concerns. Seven (39%) caregivers were male, 16 (89%) were spouses of the PwPD and median ZBI (short version) score of caregivers was 19 (IQR 12); a score of 17 or greater indicates high caregiver burden [54]. Eight PwPD opted to be interviewed with their caregivers, five of these had a diagnosis of cognitive impairment/ dementia. Therefore 29 interviews took place including 38 participants (20 PwPD, 18 caregivers). Interviews lasted between 18 and 61 minutes (mean 37 +/- standard deviation 10).

### Findings

Four distinct themes were developed that provided a descriptive coherent representation of the data (Table 3).

### Theme one: Struggling with thoughts and feelings about falling

**Different aspects of falls trigger varying emotional responses.** Many different aspects of experiencing falls provoked troubling thoughts and feelings. Emotional responses also

**Table 1. Demographic details of people with Parkinson's disease.**

| Characteristic | N (%) or Median (IQR; range) |
|---|---:|
| **Gender** | |
| • Male | 14 (70%) |
| • Female | 6 (30%) |
| **Age** | 72.5 (8.8; 57–85) |
| **Duration of PD (years)** | 12.8 (6.0; 2.5–12.5) |
| **Cognitive impairment/ dementia** | |
| • Diagnosis of cognitive impairment or dementia | 4 (20%) |
| • Self-reported/ caregiver reported concerns | 4 (20%) |
| • No diagnosis or self-reported concerns | 12 (60%) |
| **Short form FES-I score** | 13 (9; 9–25) |
| **Number of falls/ last year** | 4 (7.5; 2–100) |
| **Living situation** | |
| • With caregiver | 17 (85%) |
| • Alone | 3 (15%) |
| **Interviewed** | |
| • Alone | 12 (60%) |
| • With caregiver | 8 (40%) |

Abbreviations: FES-I = Short Form Falls Efficacy Scale International; IQR = Interquartile Range

**Table 2. Characteristics of people with Parkinson's disease and their caregivers.**

| PwPD ID | Caregiver ID | PwPD Characteristics | | Duration of PD (years) | Cognitive impairment/ dementia diagnosis | Short Falls Efficacy Scale- International (FES-I) | Number of falls / last year | Caregiver characteristics | | Joint or separate interviews |
|---|---|---|---|---|---|---|---|---|---|---|
| | | Gender | Age (Years) | | | | | Gender | Zarit Burden Interview Short Version (ZBI) | |
| PwPD 1 | Caregiver 1 | Male | 78 | 7.5 | No | 20 | 12 | Female | 27 | Joint |
| PWPD2 | Caregiver 2 | Male | 78 | 14 | No (Concerns raised) | 20 | 2 | Female | 19 | Joint |
| PwPD 3 | Caregiver 3 | Male | 85 | 2.5 | No | 11 | 2 | Female | 19 | Separate |
| PwPD 4 | Caregiver 4 | Male | 67 | 8 | No (Concerns raised) | 10 | 3 | Female | 29 | Separate |
| PwPD 5 | N/A Lives alone | Female | 73 | 18 | No | 13 | 12 | Male | N/A | N/A |
| PwPD 6 | Caregiver 6 | Female | 57 | 21.5 | No | 12 | 4 | Male | 20 | Separate |
| PwPD 7 | Caregiver 7 | Male | 73 | 11.7 | No | 20 | 4 | Female | 24 | Separate |
| PwPD 8 | Caregiver 8 | Female | 73 | 19 | Cognitive impairment | 19 | 3 | Male | 18 | Separate |
| PwPD 9 | Caregiver 9 | Male | 77 | 12.6 | No | 12 | 20 | Female | 17 | Joint |
| PwPD 10 | Caregiver 10 | Male | 66 | 14 | Dementia | 25 | 3 | Female | 33 | Joint |
| PwPD 11 | Caregiver 11 | Male | 75 | 10 | No (Concerns raised) | 9 | 4 | Female | 34 | Separate |
| PwPD 12 | Caregiver 12 | Female | 65 | 18 | No | 11 | 4 | Male | 12 | Separate |
| PwPD 13 | N/A Lives alone | Female | 70 | 3 | No | 15 | 4 | Male | N/A | N/A |
| PwPD 14 | Caregiver 14.1, 14,2* | Male | 69 | 14.9 | No (Concerns raised) | 24 | 10 | Female, Female | 48 | Joint |
| PwPD 15 | Caregiver 15 | Male | 84 | 15 | Dementia | 9 | 100 | Male | 29 | Joint |
| PwPD 16 | Caregiver 16 | Male | 72 | 5.1 | Cognitive impairment | 15 | 5 | Female | 17 | Joint |
| PwPD 17 | Caregiver 17 | Male | 79 | 16.1 | No | 11 | 100 | Female | N/A | N/A |
| PwPD 18 | Caregiver 18 | Female | 59 | 12 | No | 22 | 3 | Male | 18 | Separate |
| PwPD 19 | Caregiver 19 | Male | 69 | 9.8 | No | 11 | 4 | Female | 0 | Joint |
| PwPD 20 | Caregiver 20 | Male | 71 | 13 | No | 13 | 6 | Female | 6 | Separate |

* One PwPD had two caregivers (wife and daughter) who were both interviewed. Only caregiver 14.1 completed the ZBI short version (PwPD's wife)

Abbreviations: FES-I = Short Form Falls Efficacy Scale International, possible scores 7–28, 7–8 = low concern, 9–13 = moderate concern, 14–28 = high concern of falls [58]; ZBI = Zarit Burden Interview Score (short version), possible scores 0–48, score of 17 or greater indicates high caregiver burden [54].

changed somewhat over time. When falls first occurred PwPD described feeling unprepared, shocked, and frustrated, even if they had received information about falls or heard about falling in other PwPD.

> "*It was like falling off the edge of a cliff. . . you know realistically that you're in for a downhill run, but you expect it to be a gentle slope*".

*Caregiver 11*

Falls could also lead to embarrassment and a few PwPD, all men, had concealed falls from their caregiver. Falls reduced confidence and heightened worry, which was often most apparent immediately after a fall. Most PwPD and caregivers were fearful of falls-associated injuries. Many PwPD and caregivers were concerned about the PwPD's inability to get up from the

Table 3. Themes and subthemes.

| Theme | Subtheme |
|---|---|
| Theme one: struggling with thoughts and feelings about falling | Different aspects of falls trigger varying emotional responses |
| | Dealing with thoughts and feelings about falls |
| Theme two: recognising and Managing Risks surrounding falling | Striving to understand falls |
| | Making behavioural and practical adaptations |
| | Living a more limited life because of adaptations |
| Theme three: navigating health and care provision for falling | Frustration with inadequate information, care, and support |
| | Failing to engage with available falls information and support |
| Theme four: changing as a couple due to falling | Friction in relationships |
| | From partner to caregiver/manager |

floor after a fall, and PwPD had often previously required physical or verbal assistance, most commonly from their caregiver. Most caregivers, including all of those of a PwPD with cognitive impairment or dementia, discussed the importance of staying calm and thinking methodically after a fall, which helped reduce caregiver anxieties about why a fall had occurred or how to get the PwPD up. When providing physical assistance, both male and female caregivers often felt they were at risk of injury, and a few female caregivers experienced worry about their inability to lift the PwPD.

"*Lifting my husband did put me at more risk. . . if I was injured, I wouldn't be of so much use. . .. I'm aware of the risks but I can't make go away*"

*Caregiver 10*

Adjustment of PwPD and caregivers' activities in response to falls led to feelings of loss, frustration, and isolation in both PwPD and caregivers. These emotions were more frequently voiced by caregivers than by PwPD, on behalf of themselves or the PwPD. Some PwPD and caregivers found using medical equipment to reduce fall risk provoked strong negative thoughts. Perceptions included walking aids being designed to be for older people, which led to reduced use, and a dislike of adaptations to their home, in particular those safeguarding for a future decline in mobility.

"*If, heavens above, I have to go into a wheelchair or something like that the things are already in place which I resent slightly but I realise that you know it has to be done.*"

*PwPD 12*

Most PwPD and caregivers described concerns surrounding the progression of PD and falling, however, a few caregivers preferred not to think about what the future might hold as it was unmodifiable. Most PwPD and caregivers described an acceptance of falling that had developed over time, and in some, particularly where falls were more frequent, falls were normalised.

"*I tripped over that more times than he has.*"

*Caregiver 7*

**Dealing with thoughts and feelings about falls.** Participants described an array of coping strategies that they had developed in response to falling. Seeking social support was a prominent response with most PwPD and their caregivers seeking and finding support within the dyad. A few PwPD described difficulties when interacting outside of the dyad, often arising from friends' limited understanding of PD, which could be overcome by attending a local Parkinson's group or carers group.

"*[My friend's] perception of me as having Parkinson's. . .she wanted to hold me, she wanted to mother me, smother me.*"

*PwPD 24*

Some PwPD and caregivers utilised distancing and distraction to escape thinking about falls. These participants were predominantly male PwPD who had often experienced significant falls-related injuries.

"*I don't like to think about [falling]. I get wrapped up in the sport on the telly or something.*"

*PwPD 14*

These PwPD often described falls as frustrating and unpreventable. Whilst a few others found it helpful to release their pent-up emotions, which could be encouraged by their spouse or by HCPs, and a minority of caregivers used humour as a coping strategy.

"*Get it out instead of holding it in, the frustration and the bitterness and the anger. . .. we went to the psychologist*"

*Caregiver 18*

## Theme two: Recognising and managing risks surrounding falling

**Striving to understand falls.** PwPD and caregivers often sought to analyse incidents of falling to identify their aetiology. The heterogeneous nature of PD was reflected; perceived reasons included freezing, altered balance, multitasking, impaired concentration, fatigue and dyskinesia. Many perceived falls as multifactorial. Most PwPD and caregivers described environmental contributors to falls, including uneven ground, steps, doorways and small or crowded spaces. A minority of PwPD attributed falls to mis-stepping or tripping. Many PwPD and caregivers discussed variability of fall risk related to time of day or medication.

"*If her tablets have worn off . . .. she's more prone to be falling.*"

*Caregiver 6*

Some PwPD and caregivers perceived the PwPD did not always acknowledge risk factors or make appropriate alterations to their behaviour to prevent falls, which was especially strong in caregivers of a PwPD with cognitive impairment/ dementia.

"*His feet are too close together, if only he could get his legs apart. . . It's feasible. It's a bit disappointing*".

*Caregiver 7*

PwPD and caregivers were often uncertain about reasons for falling, and some perceived falls to be unpredictable. Whilst most appeared at ease with these unknowns, others voiced confusion and frustration. Some participants, particularly caregivers, perceived the risk of falling to be unrelenting, which led to anxiety.

*"I can't predict when it's going to happen, that's what I find very frustrating. . . I have a great day, I can have a bad day, and I can't see the difference between the two days."*

*PwPD 4*

**Making behavioural and practical adaptations.**   After a fall, dyads tended to reflect on events and both parties instigated changes to manage subsequent risk. Most PwPD and caregivers instigated behavioural adaptations to reduce subsequent risk, which included ensuring that PwPD took care to avoid over-reaching or carrying items whilst walking and took time when standing. Caregivers often helped to prevent and manage freezing episodes through approaches that were often self-taught, which included ensuring PD medication was taken on time and reciting specific phrases.

*"Say to him 'stop, stand still. Think what you're doing. Long slow stride'. It gets him going again."*

*Caregiver 1*

Others described managing or limiting the PwPD movements and one went so far as to '*watch every step*' [caregiver 1] the PwPD took. Where the PwPD had cognitive impairment or dementia, caregivers often instigated changes on behalf of the PwPD.

*"If he's been lying down, I give him time to acclimatise rather than just standing him up straightaway. . .. And always make sure that he's standing steadily before moving".*

*Caregiver 10*

Outside the home changes included taking care on uneven ground and adjusting routes to enable the PwPD to continue activities they enjoyed. Many PwPD and caregivers adapted to the variability of PD symptoms through altering the timing and type of activity, using a different mobility aid, and increased caregiver support when perceived to be at greater risk.

*"To go out walking, I'd need to be at the right stage in my medicines. . .when they're wearing off, I get more unsteady."*

*PwPD 6*

Practical adaptations inside the home and home modifications were predominantly driven by caregivers; keeping doors open and making space by reducing clutter. Most caregivers and a few PwPD described equipment which helped PwPD to feel more secure and less susceptible to falls. Mobility aids often helped PwPD to feel safe, maintain their independence and to continue activities outside of the home. Many altered the aid used in response to the environment, intensity of the physical activity or variability of PD symptoms.

*"We keep a wheelchair in the car, though she hasn't used it for ages, because if she freezes with Parkinson's then we're stuck."*

*Caregiver 8*

**Living a more limited life because of adaptations.** PD and falls transformed the lives of PwPD and caregivers. Activities were adapted or PwPD switched to alternatives where they felt more confident. However, many settings were perceived as non-modifiable, leading to avoidance and an increasingly sedentary and limited lifestyle.

"*I am a Cathedral guide, or I was. . . I can no longer do that because there's a risk of falling in the Cathedral. I'm sad not to be doing it.*"

*PwPD 5*

Arising from concerns about the PwPD falling whilst alone, caregivers often reduced the duration and frequency of their own outings; younger caregivers altered their work commitments, others went out when the PwPD was asleep. Others felt too uncomfortable to leave the PwPD alone and sought to undertake activities alone to maintain independence and confidence and reduce isolation.

"*She just has a little look round the shops. . . I encourage her to do that . . . if she doesn't do it on her own then she will lose all confidence. . . . it's good for her.*"

*Caregiver 12*

## Theme three: Navigating health and care provision for falling

**Frustration with inadequate information, care and support.** Some PwPD and caregivers had only discussed their falls-related worries with their spouse and appeared lost and unsure of where they might seek support. Some PwPD and caregivers were frustrated with current healthcare provision and participants commonly perceived communication between HCPs to be poor. These difficulties predominated in dyads where the PwPD had been diagnosed for longer and where the PwPD had cognitive impairment/ dementia.

"*The organisation is chaotic. . . trying to get information or be told what might happen next . . .nothing seems to be joined up, nobody seems to know what anybody else is doing.*"

*Caregiver 11*

Other inadequacies included the short duration of HCPs' appointments and limited physiotherapy provision. In response to a delay in therapy assessment or reduced perceived effectiveness of advice received from HCPs, dyads had sought private healthcare support and self-initiated changes within the home. A few were concerned about their GP's involvement, perceiving them to lack PD knowledge. Dyads, particularly those where PwPD had cognitive impairment/ dementia, also encountered difficulties with social services; support was not always available at times required or was not able to meet their needs or expectations.

"*We phoned one of Care Agencies . . . [my husband] said 'well actually what I really want is someone to help me up and down the stairs' . . . 'ah! For health and safety reasons we don't send anyone to do that'.*"

*Caregiver 11*

Most participants, caregivers more than PwPD, described positive experiences from Parkinson's UK support groups. Both caregivers and PwPD found it useful to speak with others in

a similar situation. However, most were unsure whether falling had been discussed and few were able to recall conversations with peers. Some groups provided services, such as physiotherapy and talks about falls from experienced speakers which were generally perceived as beneficial.

> "*We usually try and learn from each other. . . awkward for the consultants to give 100% advice . . . everybody's so different.*"
>
> *Caregiver 8*

Most participants had informed doctors or their Parkinson's disease nurse specialist (PDNS) of the PwPD's falls. Those that had not tended to perceive HCPs' roles as limited to medication management and/or perceive help-seeking as futile because they believed that falls were unpreventable. Amongst some caregivers of PwPD with more advanced PD or with cognitive impairment/ dementia, there was a sense that they were managing PD and falls alone.

> "*I don't believe [the consultant's] got anything to contribute. . .they've got to the end of the road. . . It's in my own hands now.*"
>
> *Caregiver 7*

A few participants described how specialist doctors and PDNS had supported them by signposting to services including physiotherapy or occupational therapy. Caregivers and PwPD were often more positive about physiotherapist and occupational therapist's roles. They described physiotherapists providing movement and behavioural strategies, which included teaching the PwPD how to get up from the floor, strategies to overcome freezing, how to rise from a chair and exercises for them to practice. Participants described occupational therapists giving them advice and equipment, which they perceived had reduced their risk of falls.

> "*We've been going to exercises cos they teach you to get up. . .they definitely help.*"
>
> *Caregiver 1*

**Failing to engage with available falls information and services.** A few PwPD and caregivers reported not receiving any verbal or written information about falls from HCPs; they were often ambivalent and perceived that HCPs had nothing to offer. However, most participants had an appetite for additional resources for falls management but were unsure of what might help. Few had self-sought falls information, and those that had linked their research efforts to their previous life roles (e.g. a retired HCP). Some participants were concerned that additional information might cause anxiety (see subtheme 'different aspects of falls trigger varying emotions).

> "*I don't want to know. . . I'm burying my head in the sand, like an ostrich I suppose*".
>
> *Caregiver 15*

In some PwPD and caregivers there was a sense they collected information about PD, including about falls. Information was often filed away, and some described difficulties engaging and absorbing information. Where information was read, this was often at a time when it was not relevant, and information was not taken in. Furthermore, it was not always apparent that materials were relevant to caregivers.

"*I've not particularly, necessarily read [falls information leaflets] . . . thought that they're for people with the condition.*"

*Caregiver 18*

Strategies communicated from HCPs were also not always implemented. In particular interventions were deemed less relevant in those with advanced PD, and in those with cognitive impairment/ dementia, who often encountered difficulty with physiotherapy strategies.

"*Very difficult now to encourage [my husband] to do proper exercises.*"

*Caregiver 10*

Getting support during or shortly following a fall was also problematic, a few female caregivers with high caregiver burden, and from dyads where the PwPD had cognitive impairment/ dementia, described calling paramedics as a last resort. Concerns included misperceptions that helping the PwPD up from the floor was not the paramedic's role ("*[Paramedics] have got. . .lifesaving things to do*" Caregiver 27), or that calling 999 would automatically lead to hospital admission. These caregivers had sometimes managed alone in situations where external support would likely have been safer and beneficial, leading to concern in family members.

## Theme four: Changing as a couple due to falling

**Friction in relationships.**   The relationship within the dyad had often changed and some described new friction relating specifically to the consequences of falls or falls avoidance. Caregivers had often acquired domestic responsibilities; this led some PwPD to feel uncomfortable as they still felt able to do the activities themselves or were unhappy with the caregiver's standards. Occasionally caregivers perceived the PwPD was overly reliant upon them. Some caregivers described encouraging the PwPD to seek support from HCPs, and many caregivers of PwPD with cognitive impairment/ dementia were frustrated when the PwPD did not follow their advice or advice from HCPs.

"*He's not really taken [physiotherapist advice] on board. . . it's a great pity.*"

*Caregiver 7*

However, two caregivers of PwPD with dementia, described that their understanding of the difficulties experienced by the PwPD had increased over time.

"*I try to consciously mentally step back and remember that [he] has problems. . .I've learnt a high degree of patience.*"

*Caregiver 15*

Despite friction and frustrations being common, a few dyads had experienced some positive changes as they spent more time with each other through adapting, adjusting, or switching to new activities.

"*I try to spend more time with [PwPD] and we can do things. . . we go ballroom dancing in the afternoon.*"

*Caregiver 6*

**From partner to caregiver/ manager.** Caregivers often appeared distressed when discussing their caregiver role. These feelings were not exclusively falls-related. Fall-related caregiving challenges included supporting the PwPD up from the floor, practical and behavioural adaptations to manage falls and restrictions on the caregiver's activities related to concerns of the PwPD falling when alone (see subthemes 'differing aspects of falls trigger varying emotions', 'making behavioural and practical adaptations' and 'living a more limited life because of adaptations'). A few described unease associated with the transition from close friend/ spouse to 'caregiver'.

"*I like to think of myself as the husband. . .it is difficult. . . I don't like the term carer. . .I don't feel that I'm in control of my time. It's like almost having a permanent, a full-time job.*"

*Caregiver 18*

Many caregivers described considerable efforts to try to maintain their pre-existing social activities, identity and/or relationships outside of the dyad. Friends, family, and neighbours also supported the PwPD, which could be pre-organised or 'ad hoc', which could facilitate caregivers to have time alone.

"*[I'm] on edge all the time wondering. . . is he going to fall? We have a sitter twice a week and I feel happy going out. . . otherwise, I manage all the time.*"

*Caregiver 14.1*

Most caregivers, and importantly all of those from dyads where the PwPD had cognitive impairment/ dementia, appeared defined by their caregiving role, with little space for anything more. A few were frustrated that the PwPD did not always acknowledge the changes that PD had made to their life. A few caregivers, with high and low caregiver burden, described numerous concerns, with a sense they soldiered on relentlessly. Attempting to prevent, manage and deal with the consequences of falls could be burdensome, leading one participant to feel "*inadequate. . .tired because [the falling] is just constant*" [caregiver 10].

## Discussion

This study is the first to explore the perspectives of both PwPD and family caregivers of falls and of healthcare services relating to falls prevention and management and to include PwPD with dementia. Our study generated a number of novel findings which will be of particular use to clinicians and researchers developing or delivering both specialist falls-management, self-management and educational services supporting PwPD and their families.

### Principal findings

Our study highlighted the significant burden experienced by both PwPD and their caregivers not only from falls, but from preventative measures to manage them. PwPD and caregivers need help in choosing from a range of strategies to manage falls including both those that enhance psychological wellbeing and those that reduce risk (where it is modifiable). Whilst the significant role that caregivers play in the management of falls has been reported previously, the inclusion of PwPD with cognitive impairment and dementia in this study highlighted how the caregiver's role further increased when PwPD's cognition declined [34, 59]. Falling often led to considerable transformation of the lives of PwPD and caregivers, with detrimental effects on activities of daily living, independence and social isolation. Whilst caregivers often

restricted the activity of PwPD, they rarely acknowledged risk of consequential muscle wasting or increased fear of falls [12, 60, 61]. PwPD who fall and their caregivers may benefit from interventions to help them assess and risk manage situations to attain a balance between falls-related and restriction-related harms.

PwPD and caregivers highlighted awareness of the multitude of reasons for unsteadiness and falling in PD, as has been reported in previous quantitative research [62]. Contrasting previous research, most participants reported that they had discussed falling with HCPs [10, 31, 63]. However, dyads described seeking to understand what caused falls and implementing mitigation strategies themselves often without timely support and guidance from HCPs. They also independently adopted a number of coping strategies to manage falls, which appeared to improve their psychological wellbeing in the face of threat of falls. These included acceptance and normalisation but also use of distraction, avoidance and humour. However, these approaches could potentially lead to a lack of attention to risk and missed opportunities to make helpful practical adjustments and share difficulties with HCPs. Dyads need help in choosing a range of strategies to manage falls including both those that enhance psychological wellbeing and those that reduce risk (where it is modifiable).

Dyads often reported that falls-based information was not available or was not remembered and was not relevant. Previous research has highlighted that nurses can act as mediators in explaining information to patients to facilitate patient-centred care and shared decision making [64]. However, in this study, most participants were unsure of HCPs' roles in falls management and commonly perceived that their role was medicines management or finding a cure for PD. Misperceptions held by PwPD of HCPs' roles has previously been highlighted as a barrier to PwPD accessing specialist palliative care [65]. Caregivers placed physical risk upon themselves, and our study highlighted that caregivers did not feel that falls management was a paramedic's role: reluctance to ask for support from HCPs has been reported elsewhere [34, 59]. This finding may in part be explained by advancing participant age; whilst not identified in our study, older individuals have previously described HCPs as authoritative figures who are not to be questioned [64, 66]. HCPs should seek to provide dyads with falls-information at a time when it is deemed relevant, to enhance the likelihood of it being read and facilitate shared-decision making. Dyads require information about different HCP's roles, reassurance that help-seeking is legitimate and support in planning for who to contact and when.

Many participants described difficulties accessing healthcare to meet their needs. Common difficulties included limited appointment frequency and duration, poor communication between HCPs and long waiting times for referrals between HCPs, all of which have been reported previously in PD, but not specific to falls [63, 65, 67]. Paid support was not always able to meet PwPD's individualised needs, particularly PwPD with cognitive impairment/ dementia, leaving caregivers no option but to struggle on regardless. PwPD and caregivers frequently presented as a team and worked together to navigate falls and falls management. Whilst the shifts of roles and responsibilities in dyads and the transition of spouse to caregiver has been reported previously, the inclusion of PwPD with and without cognitive impairment/ dementia in our study illustrated that with disease progression and cognitive decline the caregiver's role dominates [51].

Dyads displayed variety in their experiences and unmet needs for successful falls management. Previous research has identified that the provision of relevant, patient-centred information can enhance empowerment and engagement of patients without PD to make informed decisions about their treatment goals and preferences [64, 68–70]. In this study, cognitive impairment/ dementia was associated with additional fall risk, as has been described previously [10, 11]. Elsewhere, individuals with cognitive impairment but without PD have described falls prevention advice as irrelevant, and their caregivers have described greater

concern about falls than the individual who falls [23]. Our study built upon this concept; caregivers of PwPD with cognitive impairment/dementia described how they had instigated changes to mitigate falls on behalf of the PwPD, were frustrated when the PwPD did not follow advice from themselves or HCPs and were more likely to describe being defined by their caregiver role. Interventions provided by HCPs were also deemed less effective and relevant in the setting of cognitive impairment/ dementia. Given the consequences of cognitive impairment on falls risk, and added burden associated with falls prevention identified in this study, it is vital that future falls- based interventions recognise the significance of cognitive decline. Whilst therapy interventions may be less effective at preventing falls in those with more advanced PD, who are also more likely to have cognitive impairment/ dementia, it is imperative that caregivers are provided with appropriate advice and support to enhance their physical and psychological well-being [22]. A meta-analysis of interventions to support caregivers of older adults without PD, including individuals with dementia, concluded that interventions including those based on psychotherapy, psychoeducation, caregiver training and respite care provision, led to improvements in caregiver knowledge, burden, depression and subjective well-being [71].

Enhanced patient-professional communication, and involvement of PwPD in their care has previously led to improvements in treatment compliance, clinical outcomes and patient satisfaction [66, 72]. However, in this study, many barriers were identified to effective communication. The multi-disciplinary team was often perceived as fragmented, which has previously been identified as a barrier to patient-centred care in older people in general, PwPD without falls and in older people with dementia who fall [64, 67, 68, 73]. Dyads frequently encountered difficulties navigating the healthcare system; exploring the healthcare system from HCPs' perspective might provide a more comprehensive overview of the difficulties described. Educating PwPD and caregivers of HCPs roles, and encouraging them to share difficulties with HCPs, may improve patient- professional communication and facilitate shared decision making [74].

## Strengths and limitations

Contrasting many previous falls studies, we utilised qualitative methodology to prioritise PwPD and caregiver perspectives. This study was conducted by a multi-disciplinary team of researchers including doctors specialising in the care of PwPD who fall and a psychologist with expertise in qualitative research and emotional aspects of neurodegenerative disorders, facilitating a deeper contextualised understanding of our findings. A fairly large and varied sample of participants were included, and we were successful in sampling PwPD with cognitive impairment/ dementia and caregivers, who have been excluded from previous research [37–44]. Conducting dyadic interviews stimulated participants' thoughts; concepts were discussed that might not have been remembered, and sharing and contrasting of ideas enhanced the depth of the data [37, 55]. Dyadic interviews also facilitated the inclusion of PwPD with cognitive impairment/ dementia.

The recruitment strategy may have attracted those who have strong views about falling and its management or those who were more proactive in falls management, as these individuals might have been more forthcoming to take part. Only PwPD who attend Parkinson's UK support groups were represented. Clinical data was obtained from PwPD through self-report. No demographic data was collected from caregivers. More research is needed to elucidate how PwPD's clinical characteristics and caregiver demographics might influence the issues reported here. Whilst no ethnicity or deprivation data was collected, the sample appeared predominantly Caucasian and middle-class. Falls-associated challenges and dyads interactions

with HCPs might be different (worse) in individuals who are non-Caucasian or are less socially advantaged [75, 76].

Difficulties interviewing PwPD included hypophonia (quiet voice) and reduced recall secondary to cognitive impairment/ dementia. Therefore, it was important to allow PwPD to be interviewed with a caregiver, eight PwPD opted for this. However, the disadvantage of joint interviews is the risk of caregiver domination (which we noted sometimes) and discomfort discussing difficulties/ talking freely (which we did not note but acknowledge is possible).

## Conclusion

Caregivers and PwPD reported difficulties accessing information and were unsure of the role of HCPs in falls and when to seek help. The inclusion of caregivers for PwPD demonstrated that caregivers are key to implementing falls interventions and choosing strategies to mitigate and cope with falls, often with little input from HCPs. The caregiver burden was worse amongst caregivers of PwPD with cognitive impairment/ dementia. It is important that these difficulties are recognised by researchers and HCPs, who should seek to improve dyads' awareness of HCPs' roles, promote patient- professional communication and recognise the enhanced difficulties encountered by dyads in the setting of cognitive impairment/ dementia.

## Supporting information

**S1 File. Interview schedule.** Semi-structured interview schedule for PwPD and caregivers. (DOCX)

## Acknowledgments

The views expressed are those of the authors.

## Author Contributions

**Conceptualization:** Charlotte L. Owen, Helen C. Roberts.

**Data curation:** Charlotte L. Owen, Christine Gaulton.

**Formal analysis:** Charlotte L. Owen, Laura Dennison.

**Funding acquisition:** Charlotte L. Owen, Helen C. Roberts.

**Investigation:** Charlotte L. Owen, Christine Gaulton.

**Methodology:** Charlotte L. Owen, Helen C. Roberts, Laura Dennison.

**Project administration:** Charlotte L. Owen.

**Supervision:** Helen C. Roberts, Laura Dennison.

**Validation:** Charlotte L. Owen, Laura Dennison.

**Visualization:** Charlotte L. Owen.

**Writing – original draft:** Charlotte L. Owen, Helen C. Roberts, Laura Dennison.

**Writing – review & editing:** Charlotte L. Owen, Christine Gaulton, Helen C. Roberts, Laura Dennison.

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
