## [Decision Letter · Decision Letter 0]

9 May 2022

PONE-D-21-25994Perceptions of people with Parkinson’s and their caregivers of falling and falls-related healthcare services- a qualitative studyPLOS ONE

Dear Dr. Dennison,

Thank you for submitting your manuscript to PLOS ONE. After careful consideration, we feel that it has merit but does not fully meet PLOS ONE’s publication criteria as it currently stands. Therefore, we invite you to submit a revised version of the manuscript that addresses the points raised during the review process.

We look forward to receiving your revised manuscript.

Kind regards,

Daswin De Silva

Academic Editor

PLOS ONE

Journal Requirements:

Reviewers' comments:

Reviewer's Responses to Questions

**Comments to the Author**

1. Is the manuscript technically sound, and do the data support the conclusions?

Reviewer #1: Yes

Reviewer #2: Yes

2. Has the statistical analysis been performed appropriately and rigorously? 

Reviewer #1: N/A

Reviewer #2: Yes

3. Have the authors made all data underlying the findings in their manuscript fully available?

Reviewer #1: No

Reviewer #2: Yes

4. Is the manuscript presented in an intelligible fashion and written in standard English?

Reviewer #1: Yes

Reviewer #2: Yes

5. Review Comments to the Author

Reviewer #1: Thank you for the opportunity to read you well written and interesting paper that deals with the important topic of falling and people with Parkinson's disease. This paper addresses a novel perspective, exploring the experiences of participants, including those people with cognitive impairment / dementia and their carers. I only have a few minor comments to make that most likely need addressing:

- line 87: spell out HCP as first time use of abbreviation

- line 90: possible change "in the setting" to "context"

Methods

Your paper needs to describe in more detail. with justification, the methodology you used to address your research questions and have a section on research team reflexivity

Who developed, and how (based on what), the interview questions. I think the key questions should be in the body of the paper (not in a supplementary file) as they inform the nature of the data collected

What experience or training did the interviewers have in qualitative interviewing?

When were field notes collected and what sort of field notes were collected?

Fra more detail of what methodological approach you used in your data analysis and a more details description of what you actually did in this analysis.

Lines 161-162: Why after 18 interviews had been coded ... and not 19 or 20 or ...

In gaining consent from caregivers for those participants who had cognitive impairment/dementia, what did you do to affirm that the person agreed with their involvement - did you check with them in some way?

Results

You should not start a sentence with a number, it should be spell out.

Please state how many people volunteered, how many were ineligible etc

Your findings and discussion were well presented

Reviewer #2: This is an important topic that has not been adequately researched. The qualitative approach with both PwPD and the caregiver was most appropriate and the analysis of the data was sound. Strengths and limitations of the study were appropriately acknowledged.

6. PLOS authors have the option to publish the peer review history of their article (what does this mean?). If published, this will include your full peer review and any attached files.

Reviewer #1: **Yes: **Leigh Hale

Reviewer #2: No

---

## [Author Response · Author response to Decision Letter 0]

20 Aug 2022

Thank you for providing feedback about the journal’s requirements. We have addressed these as described below and have also noted any consequential changes to the manuscript in the table of changes that follows the cover letter.

Journal Requirements:

Thank you for this feedback. We have re-read the style requirements in each of the PDFs quoted and are unable to identify any areas where our manuscript does not meet these. Correspondence from yourselves on 24th June 2022 advised that this text is included in all revision decision letters to ensure that PLOS style is adhered to. We feel that our manuscript is in line with PLOS style guidelines and have therefore not made any additional edits here. Please advise us if you do identify any changes that we need to make and we will implement these.

Dr Charlotte Owen was supported by the University of Southampton NIHR Academic Clinical Fellow (ACF) training programme (https://www.nihr.ac.uk/funding/nihr-academic-clinical-fellowships-in-medicine-2021/25719), and subsequently went on to be awarded a fellowship by the National Institute for Health Research Applied Research Collaboration (NIHR ARC) Wessex. Work completed during the ACF training programme, and the fellowship formed the research for this manuscript. No grants were awarded for this research, we are sorry that this was unclear from the information provided. In the ‘funding information’ section within the submission system we have still listed Dr Owen’s NIHR ARC fellowship, please advise us if you would like for us to remove the fellowship listing from this section to alleviate any confusion. 

There are ethical restrictions on sharing our de-identified data set, as manuscripts contain potentially sensitive information, as imposed by the University of Southampton Faculty of Medicine ethics committee (ergo@soton.ac.uk, reference 29763). Bona fide researchers, subject to registration may request supporting data via University of Southampton repository https://doi.org/10.5258/SOTON/D2329.

We have provided this information on our cover letter alongside their contact details.

Thank you for highlighting that the corresponding author is not currently employed by the University of Southampton. We have amended the corresponding author (line 6) to Dr Laura Dennison, who is employed by the University of Southampton, and provided her email address on line 13.

Thank you for this feedback. We have reviewed all of our references and we cannot locate a reference that has been retracted. Correspondence from yourselves on 24th June 2022 advised that this text is included in all revision decision letters to ensure that PLOS style is adhered to. On reviewing our references we did identify that references 46 and 64 were the same publication (McLaughlin et al.). To rectify this we have removed reference 64 and references after reference 64 have consequently been renumbered. We have also edited our referencing style to adhere to PLOS style; citation numbers in the manuscript are now within square and not round backets.

We have outlined how we have addressed the reviewer comments in our 'Response to reviewer' document which we have uploaded to the submission system. These are also listed below. Line references are taken from the marked up copy of the manuscript.

line 87: spell out HCP as first time use of abbreviation: Line 87. Change made: Thank you for highlighting this omission. We have now spelt out ‘healthcare professionals’ on first use of the abbreviation ‘HCPs’

line 90: possible change “in the setting” to “context”. Change made: Thank you for highlighting this. We have made this suggested change on line 90.

Your paper needs to describe in more detail with justification, the methodology you used to address your research questions and have a section on research team reflexivity. Change made: Thank you for prompting us to provide a more thorough description, with justification, of the methodology used in our research. We have addressed this and included more details of our study design and methodology decisions under the subheading ‘Study Design’ on lines 120-133. Thank you also for highlighting the importance of including a section on research team reflexivity, which we have addressed through the addition of the subsection titled ‘researcher backgrounds and reflexivity’ on lines 211-228.

Who developed and how (based on what) the interview questions. I think the key questions should be in the body of the paper (not in a supplementary file) as they inform the nature of the data collected. Change made: Thank you for highlighting that we have not described who, and how, we developed our interview questions. Our questions were developed by CO, LD and HCR, as is now described within the text (lines 162-167). The interview schedule underwent minor changes during data collection as we have now described. As suggested, we have now inserted a figure which details the key interview questions (Figure 1, line 168).

What experience or training did the interviewers have in qualitative interviewing? Change made: Thank you for prompting us to describe the training that CO and CG received in qualitative interviewing. We have addressed this by describing the training and supervision that CO and CG received on lines 177-180.

When were field notes collected and what sort of field notes were collected? Change made: Thank you for highlighting that we have not described our field notes in detail. We have detailed the field notes that we collected, and how this aided our analysis, on lines 180-185.

Far more detail of what methodological approach you used in your data analysis and a more detailed description of what you actually did in this analysis. Change made: Thank you for highlighting that we needed to provide more detail about our methodological approach. We have addressed this by providing additional details to include a description of the subtype of thematic analysis that we used, how we coded our data and how we developed out subthemes and themes (lines 186-210).

Line 161-162: why after 18 interviews had been coded ... and not 19 or 20 or ... Change made: Thank you for highlighting that we had not described why we chose this timepoint. We chose this timepoint as during discussions within the research team it became clear that the number of codes was becoming unwieldly, and that many of the current codes could be grouped to support the analysis of the data. This is now described within the manuscript (lines 201-205).

In gaining consent from caregivers for those participants who had cognitive impairment/dementia, what did you do to affirm that the person agreed with their involvement - did you check with them in some way? Change made: Thank you for highlighting this important ethical consideration. We asked both PwPD and their caregivers whether they wanted to participate. Consultee declaration forms were completed when the PwPD wished to participate but lacked the capacity to provide informed written consent. We have added these additional details to the manuscript (lines 231-233).

You should not start a sentence with a number, it should be spell out. Change made: Thank you for raising this grammatical error. Due to the changes that we have made in response to the reviewer comment below, the sentence no longer starts with ‘38’

Please state how many people volunteered, how many were ineligible etc. Change made: Thank you for highlighting that we had not provided details of the number of people who were eligible to take part in the study. We have addressed this by detailing the number of questionnaires distributed and returned. All PwPD invited to interview using the purposive sampling technique agreed to participate. (lines 147-158). We have added these details into the methods section to eliminate overlap between the methods and results sections.

---

## [Decision Letter · Decision Letter 1]

11 Oct 2022

Perceptions of people with Parkinson’s and their caregivers of falling and falls-related healthcare services- a qualitative study

PONE-D-21-25994R1

Dear Dr. Dennison,

We’re pleased to inform you that your manuscript has been judged scientifically suitable for publication and will be formally accepted for publication once it meets all outstanding technical requirements.

Kind regards,

Daswin De Silva

Academic Editor

PLOS ONE

Additional Editor Comments (optional):

Reviewers' comments:

Reviewer's Responses to Questions

**Comments to the Author**

1. If the authors have adequately addressed your comments raised in a previous round of review and you feel that this manuscript is now acceptable for publication, you may indicate that here to bypass the “Comments to the Author” section, enter your conflict of interest statement in the “Confidential to Editor” section, and submit your "Accept" recommendation.

Reviewer #1: All comments have been addressed

Reviewer #2: All comments have been addressed

2. Is the manuscript technically sound, and do the data support the conclusions?

Reviewer #1: Yes

Reviewer #2: Yes

3. Has the statistical analysis been performed appropriately and rigorously? 

Reviewer #1: N/A

Reviewer #2: Yes

4. Have the authors made all data underlying the findings in their manuscript fully available?

Reviewer #1: (No Response)

Reviewer #2: Yes

5. Is the manuscript presented in an intelligible fashion and written in standard English?

Reviewer #1: Yes

Reviewer #2: Yes

6. Review Comments to the Author

Reviewer #1: (No Response)

Reviewer #2: I had actually accepted the article previously so only the other reviewers comments really needed to be addressed.

7. PLOS authors have the option to publish the peer review history of their article (what does this mean?). If published, this will include your full peer review and any attached files.

Reviewer #1: **Yes: **Leigh Hale

Reviewer #2: No

---

## [Editor Report · Acceptance letter]

13 Oct 2022

PONE-D-21-25994R1 

Perceptions of people with Parkinson’s and their caregivers of falling and falls-related healthcare services- a qualitative study 

Dear Dr. Dennison:

I'm pleased to inform you that your manuscript has been deemed suitable for publication in PLOS ONE. Congratulations! Your manuscript is now with our production department. 

Kind regards, 

on behalf of

Dr. Daswin De Silva 

Academic Editor

PLOS ONE